# Facilitating Informed Decision Making: Determinants of University Students’ COVID-19 Vaccine Uptake

**DOI:** 10.3390/vaccines10050704

**Published:** 2022-04-29

**Authors:** Tugce Varol, Francine Schneider, Ilse Mesters, Robert A. C. Ruiter, Gerjo Kok, Gill A. Ten Hoor

**Affiliations:** 1Department of Work and Social Psychology, Maastricht University, 6200 MD Maastricht, The Netherlands; r.ruiter@maastrichtuniversity.nl (R.A.C.R.); g.kok@maastrichtuniversity.nl (G.K.); gill.tenhoor@maastrichtuniversity.nl (G.A.T.H.); 2Department of Health Promotion, CAPHRI, Maastricht University, 6200 MD Maastricht, The Netherlands; francine.schneider@maastrichtuniversity.nl; 3Department of Epidemiology, CAPHRI, Maastricht University, 6200 MD Maastricht, The Netherlands; ilse.mesters@maastrichtuniversity.nl

**Keywords:** vaccine, COVID-19, intention, determinants, university students

## Abstract

Background: Although several COVID-19 vaccines are available, the current challenge is achieving high vaccine uptake. We aimed to explore university students’ intention to get vaccinated and select the most relevant determinants/beliefs to facilitate informed decision making around COVID-19 vaccine uptake. Methods: A cross-sectional online survey with students (*N* = 434) from Maastricht University was conducted in March 2021. The most relevant determinants/beliefs of students’ COVID-19 vaccine intention (i.e., determinants linked to vaccination intention, and with enough potential for change) were visualized using CIBER plots. Results: Students’ intention to get the COVID-19 vaccine was high (80%). Concerns about safety and side effects of the vaccine and trust in government, quality control, and the pharmaceutical industry were identified as the most relevant determinants of vaccine intention. Other determinants were risk perception, attitude, perceived norm, and self-efficacy beliefs. Conclusion: Our study identified several determinants of COVID-19 vaccine intention (e.g., safety, trust, risk perception, etc.) and helped to select the most relevant determinants/beliefs to target in an intervention to maximize COVID-19 vaccination uptake. Concerns and trust related to the COVID-19 vaccine are the most important targets for future interventions. Other determinants that were already positive (i.e., risk perception, attitudes, perceived norms, and self-efficacy) could be further confirmed.

## 1. Introduction

The world has been trying to combat the COVID-19 pandemic since late December 2019 [1]. Governments implemented public health measures that were deemed to be the only way to prevent the spread of SARS-CoV-2 until the roll-out of the COVID-19 vaccines [2,3]. However, new developments brought new challenges, such as vaccine donation (see, e.g., [4]) and COVID-19 vaccine hesitancy, defined by the WHO Strategic Advisory Group of Experts (SAGE) on Immunization as the delay in acceptance or refusal of vaccines despite availability of vaccine services [5,6,7].

Since several COVID-19 vaccines were developed or are currently under development, people’s intention to get the COVID-19 vaccine as a vital step is the focus of health professionals and governments. High vaccine uptake is deemed important to control the spread of COVID-19 [7,8]. Several studies demonstrated that people’s intention to get vaccinated against COVID-19 is positive, yet not positive enough [9,10] and that there is room for improvement. To increase vaccine uptake, identifying the so-called determinants/beliefs behind people’s intention to engage in health behavior, such as vaccination against COVID-19, is the key to develop successful evidence and theory-based interventions [11,12]. As behavior change methods do not directly operate on the behavior itself but on its determinants, intervention developers first need to map the determinants of behavior/intention and then select the most relevant ones for an intervention [12,13]. In a systematic review by Larson et al. [14], an attempt was made to understand vaccine hesitancy and its determinants but answers remained inconclusive: they concluded that determinants of vaccine hesitancy are context-specific and varying across time, place and type of vaccine. Therefore, in this study, we systematically determined and selected the most relevant determinants/beliefs of COVID-19 vaccine intention of university students.

### 1.1. Theories behind the Study

An earlier meta-analysis has shown clear support for the utility of Theory of Planned Behavior in explaining vaccine hesitancy [15]. The Theory of Planned Behavior (or in updated version the Reasoned Action Approach [16]; Theory of Planned Behavior [17]) postulates that behavior is influenced by one’s intention to engage in that behavior, and intention is influenced by three determinants with underlying beliefs: (a) attitude, one’s (positive/negative) evaluation of the consequences of engaging in a behavior; (b) perceived norm, one’s perception that important others might (dis)approve of them for engaging in a behavior (injunctive norm) and one’s perception that others like themselves do (or do not) engage in a behavior (descriptive norm); (c) perceived behavioral control (or self-efficacy), one’s perception about whether they are capable of, or have control over, executing a behavior. Protection Motivation Theory [18,19], on risk perception, declares that (a) threat appraisal, people’s perception of the severity of a threat (perceived severity) and people’s perception of how susceptible they are to a threat (perceived susceptibility), and (b) coping appraisal, people’s expectation of whether engaging in a behavior will lead to a change (response efficacy) and people’s perception of whether they can perform a behavior successfully (self-efficacy), determine people’s risk perception and how they will respond to a threat. In the case of vaccination intention, determinants related to automaticity and habit do not seem to be essential.

### 1.2. Current Study

The aims of this study are to (1) examine university students’ intention to get the COVID-19 vaccine and (2) select the most relevant (i.e., correlated to one’s intention, and showing room for improvement) determinants/beliefs of students’ intention to get vaccinated to target in an intervention. By COVID-19 vaccine, we refer to vaccines that are approved for use in the EU at the time that this study was executed.

## 2. Materials and Methods

### 2.1. Participants and Recruitment

Maastricht University students were recruited (8 March until 29 March 2021) through a student panel operated by Flycatcher (2021) (an online survey platform https://www.flycatcher.eu/en/Home/OverOns [accessed on 21 March 2022]). The student panel is refreshed at the beginning of each academic year by including new students and is representative of all the study years. All panel members of the student panel were invited to the survey. Three reminders were sent to the students (on 15, 22, and 25 March). Students who completed the survey received a small incentive for their participation. This study was approved by the Ethics Review Committee Psychology & Neuroscience, Maastricht University (reference number 188_10_02_2018_S59).

### 2.2. Design and Procedure

The cross-sectional online survey could be accessed upon clicking the hyperlink sent with an e-mail invitation. After informed consent, students received questions on the topics of (1) their views on the risk of contracting COVID-19 and its severity (risk perception); (2) concerns and trust around the COVID-19 vaccine (concerns and trust—specific attitudinal and risk-perception beliefs); (3) their opinions about getting the COVID-19 vaccine (attitude); (4) what they think about what other people will do or want them to do regarding getting the COVID-19 vaccine (perceived norm); (5) potentially difficult situations regarding getting the COVID-19 vaccine (self-efficacy); and (6) their intentions to get the COVID-19 vaccination (intention). Students were also asked about their demographic information. All questions were in English to reach all the students (both Dutch and international) within the university (note that all students have a good command of English).

### 2.3. Measurements

The questionnaire was developed based on the available literature on COVID-19-vaccine hesitancy and vaccine hesitancy in general [20,21,22,23,24] and further revised based on a qualitative pretest with students (data not published—in this pretest we asked for examples about information needs and trusted resources). The underlying theories behind the questionnaire were the Reasoned Action Approach (RAA) and the Protection Motivation Theory (PMT). Questions can be found in Appendix A.

**Intention** was assessed with the item “I intend to get the COVID-19 vaccination when invited to do so”, which was answered on a 7-point Likert scale (fully disagree (1)—fully agree (7)). Another two intention questions were asked based on two different scenarios regarding waiting to get the COVID-19 vaccine: (1) “When it is my turn, I think I will wait to see if others experience any negative side effects due to getting the COVID-19 vaccination” and (2) “When it is my turn, I think I want to wait until next year before I make a decision about getting the COVID-19 vaccination” with a 7-point Likert answer option and in case, they are not willing to get the COVID-19 vaccine, “I do not intend to take the vaccination” response option was included.

**Risk perception** was assessed with five items such as “I think that without vaccination, I might be at risk of contracting COVID-19”; “I think that if I contract COVID-19, the physical consequences for me would be severe”; and “I know people who have severe health problems due to COVID-19”. All items were answered on a 7-point Likert scale; fully disagree (1)—fully agree (7).

**Concerns and trust** is partly underlying attitude and risk perception, and focused on students’ evaluations about the development, safety, possible short- and/or long-term side effects of the COVID-19 vaccine as well as students’ trust in government, pharmaceutical industry, and quality control with regard to the COVID-19 vaccine. Additionally, three items were included to compare the COVID-19 vaccine with current vaccines in the National Immunization Program in relation to safety, effectiveness, and trustiness. There were 14 items in total; example items are “I am worried about the speed of the development of the vaccine”; “I am worried about the safety of the COVID-19 vaccine”; “I am worried about possible long-term (more than a week) negative side effects of the COVID-19 vaccine”; “I trust the government about ensuring the safety of the COVID-19 vaccine”. Except for “How many people do you know who already received the COVID-19 vaccine and had no serious complaints afterwards?” item (answer option: none (1)—many (7) and I do not know people who already received the COVID-19 vaccine), all items were responded on a 7-point Likert scale (fully disagree (1)—fully agree (7)).

**Attitude** consisted of seven items, for instance, “I think that by getting the COVID-19 vaccine, I protect myself against contracting COVID-19”; “I think that getting the COVID-19 vaccine is a way out of this pandemic”; and “I think that getting the COVID-19 vaccine is my moral duty”. All attitude items were answered on a 7-point Likert scale (fully disagree (1)—fully agree (7)).

**Perceived norm** included three items with a 7-point Likert scale answer option (fully disagree (1)—fully agree (7)): “I think that most people like me will get the COVID-19 vaccination”; “I think that my doctor/health care provider wants me to get the COVID-19 vaccination”; and “I think that most people who are important to me want me to get the COVID-19 vaccination”.

**Self-efficacy** was measured with six items, e.g., “If I would decide to get the COVID-19 vaccination, I am confident that I could get it when it is my turn”; “I feel comfortable talking to my family and/or friends about whether or not to get the COVID-19 vaccination”; and “I am confident that before I decide to get the COVID-19 vaccine, I will have sufficient information about the COVID-19 vaccine”. A 7-point Likert scale was used for the answer options (fully disagree (1)—fully agree (7)).

**Demographics** were measured by asking age, gender, study year, faculty, living condition and nationality (Dutch or international).

### 2.4. Data Analysis

Descriptive statistics were analyzed by using IBM SPSS Statistics 26, and the associations between intention and all determinants/beliefs were calculated and reported (for an example, see [25]). The Confidence Interval-Based Estimation of Relevance (CIBER, [26]) approach was used to establish the determinant/belief relevance depending on (1) the association between the intention to get the COVID-19 vaccine and determinants (e.g., risk perception) and (2) the room for improvement based on the univariate distribution of each determinant/belief. For instance, if a determinant/belief has no correlation with intention but has room for improvement, this determinant/belief would unlikely be a determinant to intervene on, whereas a determinant/belief correlated with intention and has a mean score on the middle of the scale or on the undesirable direction would be a relevant target. Therefore, combining these two types of information is necessary for establishing the determinant/belief relevance [27]. While determining the relevance of a determinant/belief, it is important to check all the available information (and context) simultaneously, where the CIBER plots help inspect the information by visualizing the univariate distribution of each determinant/belief; the correlation between behavior/determinant and determinants; confidence intervals for the mean; and confidence intervals for bivariate correlations [26]. The CIBER approach also allows intervention developers to study the determinants at a high level of specificity, i.e., sub-determinants or beliefs, that can be used in the intervention messages [27], as we did in our study. Contrary to commonly used multiple regression analysis in determinant studies which assesses the total explained variance in the dependent variable based on the determinants in the model, the CIBER approach assesses the determinant/belief relevance on the individual determinant level and postulates that the multiple regression analysis can be problematic to establish the determinant/belief relevance due to the overlap between the determinants (for details see; [26]). To create the CIBER plots, the ‘behaviorchange’ R package was used. The questionnaire, Appendix A, and non-identifiable data are available at the Open Science Framework: https://osf.io/8b7pu/ (accessed on 21 March 2022).

## 3. Results

### 3.1. Background Characteristics of the Sample

A total of 908 students were invited to the survey and 483 responded (53.2% response rate). From those, 43 incomplete responses and 2 responses with poor response quality (i.e., straight lining/patterns) were removed. Another four did not consent to participating, leading to a final sample of 434 students (47.8%). The mean age of eligible students was 22 (range: 18–42 years) (panel [based on data of UM Flycatcher student panel members] = 22; range 18–43 years). A total of 75.3% of students were female (panel = 73.3%). Dutch (51.8%) and international students were equally represented; no difference in vaccination intention was found between Dutch and International students (*M* = 6.16 for Dutch students and *M* = 6.23 for international students, *p* = 0.61). For the different underlying determinants, some determinants scored significantly different, but the mean differences for the most were small (most determinants had a mean difference <0.30, and all <0.70—Data not reported but can be found in Appendix A). Detailed background information about the sample is provided in Table 1.

### 3.2. Intention to Get the COVID-19 Vaccine

Of the 434 students, 348 (80.1%; score 6–7 [fully agree]) intended to get the COVID-19 vaccination when invited to do so (11 students fully disagreed to get vaccinated against COVID-19). The mean and median scores of students’ intention were *M* = 6.20 (1–7); *SD* = 1.44; *Mdn* (*IQR*) = 7.00 (6–7); 11% of students agreed (6–7) with the item “*When it is my turn, I think I will wait to see if others experience any negative side effects due to getting the COVID-19 vaccination*”; 3.9% agreed (6–7) with “*When it is my turn, I think I want to wait until next year before I make a decision about getting the COVID-19 vaccination*”.

### 3.3. Selecting the Most Relevant Determinants/Beliefs

Almost all determinants that were selected for this study (based on theory and earlier research [20,21,22,23,24]) [1] were correlated with the intention to get vaccinated, and [2] had potential room for improvement. With that, all items that correlated significantly with intention and have room for improvement (we defined ‘room for improvement’ as having a mean score less than 6), are potentially relevant as potential targets for future interventions. All mean, median, SD, IQR and r can be found in Appendix A.

### 3.4. Concern and Trust

Although the most belief items were significantly correlated with vaccination intention, often the correlation coefficient was relatively low, or the mean score was relatively high (see Figure 1). The determinant with high correlations and the most room for improvement was “concern and trust” (except for one item where 12.4% indicated to not know anyone who already received the COVID-19 vaccine, mean scores were between 2.86 and 5.53, and *r*’s ranged from −0.27 to 0.67), and therefore an important intervention target. Items included (1) the safety and effectiveness of the vaccine, (2) possible side effects, and (3) trust in the government, the quality control and the pharmaceutical industry. Regarding three additional items comparing current vaccines in the National Immunization Program against diseases (such as measles, pertussis, diphtheria, and other diseases) with the COVID-19 vaccine showed that participants were neutral in terms of whether the COVID-19 vaccines are equally safe, effective, and trusted (i.e., mean scores close to the middle of the scale, showing that there is room for improvement; see Figure 1).

### 3.5. Other Items That Should Be Considered as Target for Future Intervention

All risk perception items (see Figure 2), except “*I had people in my social environment who had serious negative experiences related to COVID-19*” were significantly correlated with vaccination intention (*r* ranges from 0.15–0.43). Additionally, all items scored neutral or positive and had room for improvement, making them important targets for future interventions. Attitude (Figure 3), perceived norm (Figure 4), and self-efficacy (Figure 5) items had high correlations (*r’s* ranging from 0.27 to 0.72), but also had high mean scores (*M’s* ranging from 5.13–6.08), making those determinants important targets for confirmation in interventions, but not for improvement per se.

## 4. Discussion

While reopening universities, it is vital to prepare a safe educational environment for students and staff. This includes helping students to make informed decisions about the COVID-19 vaccination. In this study, we identified the reasons (determinants/beliefs) behind students’ possible hesitancy for the COVID-19 vaccine and selected the most relevant determinants/beliefs to further improve the uptake.

Based on the findings of this study, most students (80%) intended to get the COVID-19 vaccine when it is their turn. Previous studies among university students also found relatively high willingness to be vaccinated against COVID-19 [28,29,30]. Nevertheless, people’s intention to get the COVID-19 vaccine can be further enhanced by targeting its determinants.

Earlier studies on vaccine hesitancy illustrated attitude, perceived norm, and self-efficacy as determinants of people’s vaccination intention [14,15]. What is shown in our study in the context of COVID-19 is that students have positive attitudes, perceived norms and self-efficacy in relation to the COVID-19 vaccines. This is in line with what other studies found (see, e.g., [4,31]). Additionally, risk perception was found to be a determinant of students’ vaccine intention, which was in line with the findings of previous studies [32,33,34,35,36]. Therefore, those determinants should be further confirmed in future interventions.

Our study demonstrated that the concerns about the safety and side effects of the COVID-19 vaccine, and trust in the government about the safety of the vaccine, the quality control, and the pharmaceutical industry, are the most important intervention targets to improve students’ intention to be vaccinated against COVID-19. Specifically, the possible long-term side effects and safety of the COVID-19 vaccines were the main concerns among students. This is in line with the findings from other studies in which the safety and trust were found as the most important determinants of intention to get the COVID-19 vaccine as well [10,24,37]. However, when students were asked whether the COVID-19 vaccines are equally safe, effective, and/or trusted compared to the current vaccines in the National Immunization Program, students mostly scored neutral, which might, or might not, be indicative of a general hesitancy about vaccines’ safety, effectiveness, and trustiness worldwide [33]. Future (potentially more qualitative) studies could help answering this question.

### 4.1. An Intervention to Promote Informed Decision Making

Based on the findings of this study, the most relevant determinants/beliefs behind students’ intention to get the COVID-19 vaccine are listed in Table 2. For each belief, a theoretical change method is selected that fits with the general determinant [13], for example “If I contract COVID-19, the physical consequences for me would be severe” had a mean that was relatively low, in combination with a relatively low correlation. Both should be higher (that is: ideally it is desired that people perceive COVID-19 as having severe consequences). One method for increasing risk perception (and the correlation with intention to vaccinate) is “consciousness raising” (either about the risk, or about the consequences). All methods for change have so-called parameters for effectiveness that need to be fulfilled [13], for example consciousness raising should always be combined with (an improvement in) self-efficacy. In a qualitative part of this project (data not published) we asked which aspects students wanted to get information about, and by whom. Students indicated that they preferred science-based information from content experts, supported by high-level scientific publications. Based on this study, an intervention was developed that existed of a series of videos on a special website of the university on COVID-19 directed at students. The actual form was an interview by one student with, each time, an expert. The first part was about risk perception and worries and trust, with two experts in clinical microbiology, the second part on attitudes and perceived norms with two experts in health promotion/health psychology, and the third part about perceived control was covered with clear online instructions on how, where and when to get the vaccine, especially focused on international students. More information about the intervention development and lessons learned can be found in [38].

### 4.2. Limitations

The limitations of this study include: first, rapid changes happen in terms of vaccine availability (e.g., the developments with AstraZeneca vaccine) as well as the COVID-19 regulations (e.g., relaxations in the measures) and depending on these developments and the related media coverage, the intention of students to get vaccinated against COVID-19 might also change over time. Therefore, follow-up studies at different time points might be needed to have a better view of students’ intention level and its determinants. Second, we could only assess a limited number of determinants/beliefs since longer surveys might lead to a decline in the response rate. Therefore, there might be other important determinants/beliefs that might (positively or negatively) contribute to students’ vaccine intention. Additionally, the CIBER approach is helpful in eliminating irrelevant/not changeable determinants, but selection has to be carried out carefully at all times; sometimes, for example, it is needed to create interventions to keep a specific determinant at a certain high level. Systematic or scoping reviews compiling the theories used in the studies of COVID-19 vaccination or vaccination in general might be helpful for the identification of the determinants of vaccine intention and provide a roadmap for future vaccine studies. Third, this study was conducted in the Netherlands. As countries enforced varied regulations during the COVID-19 pandemic, selected relevant determinants may differ in other countries (see also [14]). Fourth, we used an already existing student panel for the data collection. Although the student panel is representative of the university students and the response was relatively high for this study, there might some deviations in the responses.

## 5. Conclusions

In conclusion, the majority of students intended to get the COVID-19 vaccination. However, there is still some hesitation in relation to the safety and side effects of the COVID-19 vaccine as well as the trust in the government, quality control, and pharmaceutical industry, which can be addressed with scientific information from trusted sources that will assist in informed decision making. All relevant determinants/beliefs can be targeted in interventions to facilitate the COVID-19 vaccine uptake.

## Figures and Tables

**Figure 1 vaccines-10-00704-f001:**
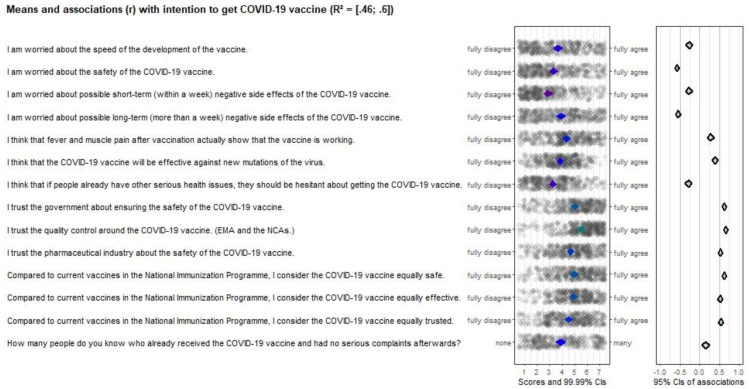
CIBER plot of concerns and trust visualizing means and association with intention to get the COVID-19 vaccine.

**Figure 2 vaccines-10-00704-f002:**
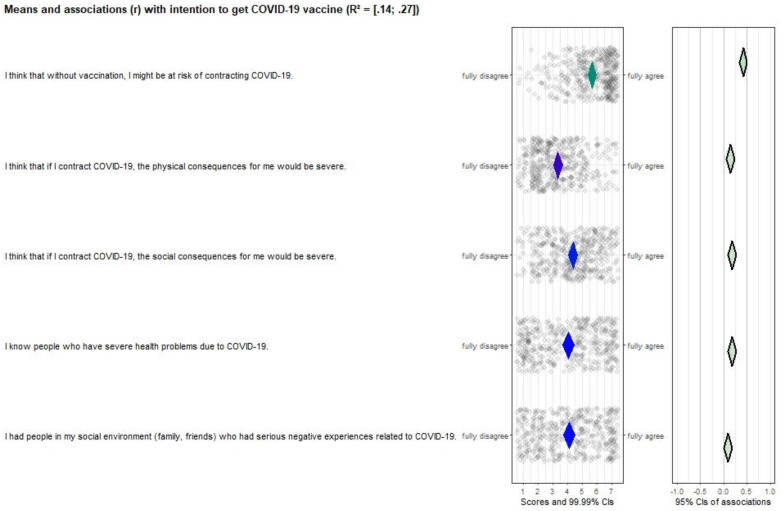
CIBER plot of risk perception visualizing means and association with intention to get the COVID-19 vaccine.

**Figure 3 vaccines-10-00704-f003:**
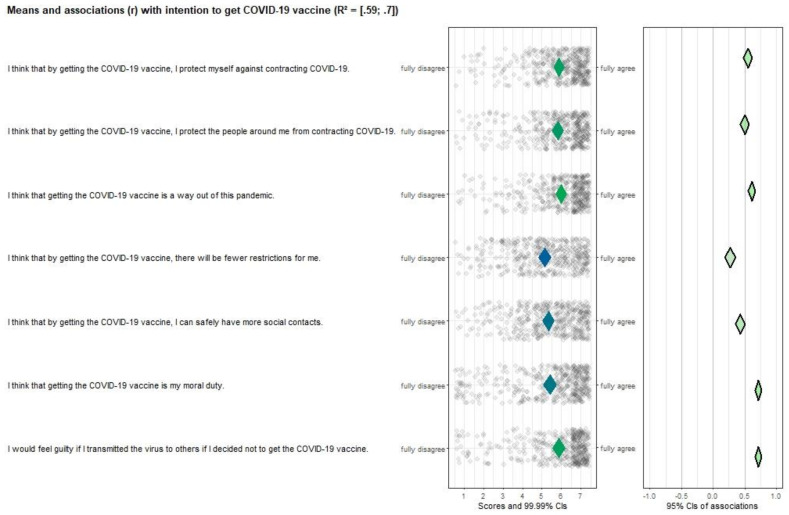
CIBER plot of attitude visualizing means and association with intention to get the COVID-19 vaccine.

**Figure 4 vaccines-10-00704-f004:**
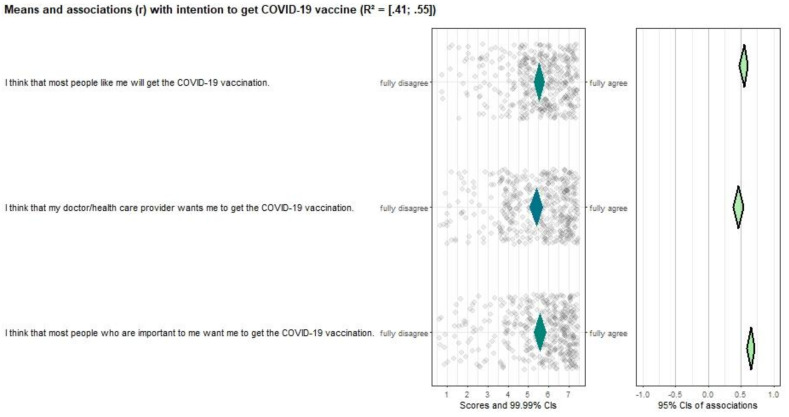
CIBER plot of perceived norm visualizing means and association with intention to get the COVID-19 vaccine.

**Figure 5 vaccines-10-00704-f005:**
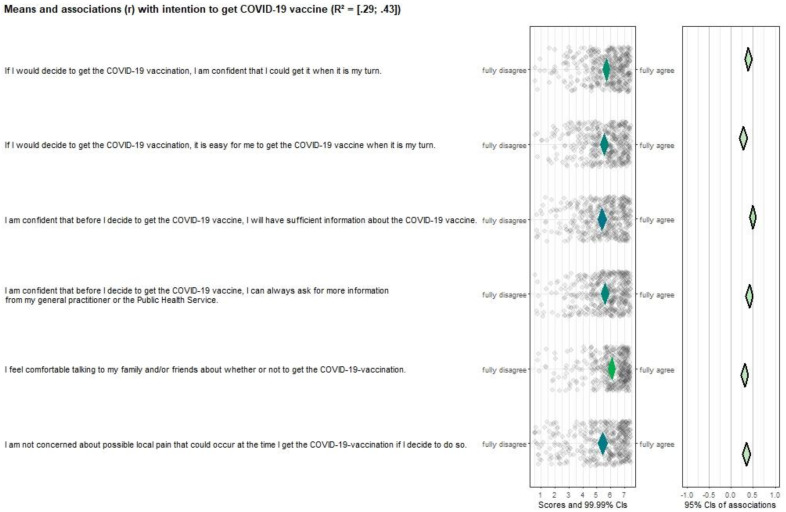
CIBER plot of self-efficacy visualizing means and association with intention to get the COVID-19 vaccine.

**Table 1 vaccines-10-00704-t001:** Background characteristics of the sample (*N* = 434).

Students	*N* (%)
** *Gender (female)* **	327 (75.3%)
** *Age in years (M + SD)* **	22.1 (3.5)
** *Study year* **	
Bachelor year 1	96 (22.1%)
Bachelor year 2	84 (19.4%)
Bachelor year 3	99 (22.8%)
Pre-master	1 (0.2%)
Master year 1	72 (16.6%)
Master year 2	51 (11.8%)
Master year 3	24 (5.5%)
Master year 4	7 (1.6%)
** *Living situation* **	
I live alone	88 (20.3%)
I live with my parent(s)/caretaker(s)	102 (23.5%)
I live with my partner	54 (12.4%)
I live with my partner and kid(s)	4 (0.9%)
I live with my kid(s)	1 (0.2%)
I live with people other than the abovementioned	185 (42.6%)
** *Faculty* **	
Faculty of Health, Medicine and Life Sciences (FHML)	178 (41%)
Faculty of Arts and Social Sciences (FASoS)	41 (9.4%)
Faculty of Psychology and Neuroscience (FPN)	50 (11.5%)
School of Business and Economics (SBE)	60 (13.8%)
Faculty of Law (FdR)	49 (11.3%)
Faculty of Science and Engineering (FSE)	56 (12.9%)
** *Nationality* **	
Dutch student	225 (51.8%)
International student	209 (48.2%)

**Table 2 vaccines-10-00704-t002:** An example of pairing the relevant determinants with behavior change principles to target in intervention.

Determinant/Item	Change *	Method	Parameters
**Risk Perception:**			
Without vaccination, I might be at risk of contracting COVID-19 If I contract COVID-19, the physical consequences for me would be severe If I contract COVID-19, the social consequences for me would be severe	5.7 ↑ 3.3 ↑ 4.4 ↑	[Belief selection/done] Consciousness raising Framing Self-affirmation	-self-efficacy improvement -gain frames -tailored to the individual
**Concerns and trust:**			
Concerns about the safety of the COVID-19 vaccine Concerns about possible long-term negative side effects of the COVID-19 vaccine The COVID-19 vaccine will be effective against new mutations of the virus I trust the government about ensuring the safety of the COVID-19 vaccine I trust the quality control around the COVID-19 vaccine I trust the pharmaceutical industry about the safety of the COVID-19 vaccine [Compared to current vaccines in the National Immunization Program:] I consider the COVID-19 vaccine equally safe I consider the COVID-19 vaccine equally effective I consider the COVID-19 vaccine equally trusted	3.3 ↓ 3.9 ↓ 3.9 ↑ 5.0 ↑ 5.5 ↑ 4.7 ↑ 5.0 ↑ 4.9 ↑ 4.5 ↑	Scenario-based risk info Persuasive communication Tailoring	-plausible cause–effect -relevant, not-discrepant, arguments -interactive (if possible?)
**Attitude/Outcome expectations:**			
By getting the COVID-19 vaccine, I can safely have more social contacts I think that getting the COVID-19 vaccine is my moral duty I would feel guilty if I transmitted the virus if I decided not to get the vaccine	5.3 ↑ 5.4 ↑ 5.9 ↑	-Shifting focus -Self-reevaluation -Anticipated regret	-new reason (postponed reward) -self-image/high self-efficacy -imagery/positive about avoiding negative consequences
**Perceived norm/Social influence:**			
People like me will get the COVID-19 vaccination My doctor/health care provider wants me to get the COVID-19 vaccination People who are important to me want me to get the COVID-19 vaccination	5.5 ↑ 5.4 ↑ 5.6 ↑	Info about others’ approval Resistance to social pressure Mobilizing social support Modeling	-are present -commitment/values -available; trust -reinforcement, identification, self-efficacy, coping
**Self-efficacy/Perceived control:**			
I am confident that I could get it when it is my turn It is easy for me to get the COVID-19 vaccine when it is my turn I will have sufficient information about the COVID-19 vaccine I can always ask for more information from my general practitioner/PHS I am not concerned about possible local pain that could occur	5.7 ↑ 5.5 ↑ 5.4 ↑ 5.6 ↑ 5.4 ↑	Modeling Guided practice Planning coping responses Goal setting	-reinforcement, identification, self-efficacy, coping -demonstration, instruction, enactment -identification and practice -commitment to the goal
**From the university:**	Advocacy/active support Technical assistance Mass-media role modeling		-matching style, consciousness raising/persuasion /approval -fit culture and resources -appropriate models being reinforced

* Numbers indicate the mean scores, and the direction of the arrow refers to the direction of the intended change.

## Data Availability

Data is available at https://osf.io/8b7pu/ (accessed on 21 March 2022).

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
