# Peer review of "Facilitating Informed Decision Making: Determinants of University Students’ COVID-19 Vaccine Uptake"

_vaccines, 2022, doi:10.3390/vaccines10050704_

Round 1

Reviewer 1 Report

The paper describes COVID-19 vaccines uptake intention and its determinants in a large sample of university students during the early period of the vaccination campaign, aiming to select the most relevant determinants for intervention development. Determinants were based on theory (referring to concepts from Theory of Planned Behavior and Protection Motivation Theory) and earlier empirical research. The relevance of the determinants (in terms of their association with intention and the room for improvement) was assessed by the CIBER approach. Possible relevant theoretical change methods are paired with important determinants.

MAJOR COMMENTS

  • I applaud the reference to the theoretical underpinnings of the study. I would appreciate a similar short description of empirical evidence (e.g., Larson et al, 2014; Xiao & Wong, 2020; Guidry et al, 2021; …) in the introduction.
    • Can the authors make reference to a theoretical account of the importance of Concerns and Trust?
  • Sample characteristics: although the student panel as a whole might be representative of all students at Maastricht University, the final sample (n=434) might not. The authors should address this in their discussion. For sure, the results are not to be generalized to the total population of Maastricht University Students (as is done on p11, line 301-302)
  • p5, lines 207: the authors might be more explicit on what is a ‘high’ correlation coefficient and what is considered ‘room for improvement’.
    • g., Results. Risk perception. Three of four significant correlations are small to medium. The strongest correlating item has limited room for improvement. Nonetheless they are all considered important targets for intervention.
  • CIBER plots. Since they are the most informative results of the study in terms of which determinant to choose for intervention, I would not place them in the Supplementary Materials. I think they are more informative than Table 2. Given the novelty of the CIBER plot approach, I feel that a thorough description of how to interpret the plot is warranted.
  • Discussion, p9, line 272. The authors state that intervention should also increase correlation with the behavior. What method do they suggest for achieving such strengthened association?

MINOR COMMENTS

  • Background characteristics. Are there relevant differences in uptake intention and determinants between Dutch and International students? I could imagine that for example trust in the government could differ. Is it known where the international students would be vaccinated (the Netherlands vs their home country)?
  • Intention. How many participants choose the “I don’t intend to take the vaccination” response option? How many answered “fully disagree”?
  • p5, lines 212-214. The additional items should be described in the method section.
  • Discussion, p8, line 243: throughout the ms, the authors used the word ‘determinant’, but now they speak of ‘predictor’. Please be consistent in your wording.
  • Discussion, p9, line 263. Do the authors have a suggestion on how to resolve whether the neutral scores on the comparison between COVID-19 and other vaccines might or might not be indicative of general vaccine hesitancy?
  • Discussion, table 3. The meaning of the Change column is unclear to me (and layout is messy). If the determinants attitude, perceived norm and self-efficacy are not considered targets for improvement (p6, line 225), why are they marked with upward arrows? (what is the meaning of the arrows?)
  • Discussion, p9, line 275-276: ‘reported elsewhere’: where?
  • Discussion, p9, intervention: Do I understand that an intervention was developed? has it been deployed/implemented/tested?
  • Discussion, limitations. The statement that there might be other determinants is vague and nugatory. Can the authors make some theory-based or empirically informed suggestions?

Author Response

#

Reviewer 1

Our response

1

The paper describes COVID-19 vaccines uptake intention and its determinants in a large sample of university students during the early period of the vaccination campaign, aiming to select the most relevant determinants for intervention development. Determinants were based on theory (referring to concepts from Theory of Planned Behavior and Protection Motivation Theory) and earlier empirical research. The relevance of the determinants (in terms of their association with intention and the room for improvement) was assessed by the CIBER approach. Possible relevant theoretical change methods are paired with important determinants.

I applaud the reference to the theoretical underpinnings of the study.

Thank you for your helpful suggestions.

2

I would appreciate a similar short description of empirical evidence (e.g., Larson et al, 2014; Xiao & Wong, 2020; Guidry et al, 2021; …) in the introduction.

Thank you for those suggestions. We agree that these are helpful additions in the introduction and have included those references:

“However, new developments brought new challenges, such as vaccine donation (see e.g. Guidry et al., 2021) and COVID-19 vaccine hesitancy, defined by the WHO Strategic Advisory Group of Experts (SAGE) on Immunization as the delay in acceptance or refusal of vaccines despite availability of vaccine services (MacDonald, 2015; WHO, 2014, p. 7).”

“An earlier meta-analysis has shown clear support for the utility of theory of planned behavior in explaining vaccine hesitancy (Xiao & Min Wong, 2020). The Theory of Planned behavior (or in updated version the Reasoned Action Approach; Fishbein and Ajzen, 2010; Theory of Planned Behavior, Ajzen, 2015) postulates”

“In a systematic review by Larson et al., (2014),  an attempt was made to under-stand vaccine hesitancy and its determinants but answers remained inconclusive: they concluded that determinants of vaccine hesitancy are context-specific and varying across time, place and type of vaccine.”

3

Can the authors make reference to a theoretical account of the importance of Concerns and Trust?

See also #2 – with adding the suggested references, this already becomes more clear. Concerns and trust are under the attitudinal and risk perception beliefs. Therefore, stated theories also apply for these items. We now further clarified this under 2.2. “Design and Procedure”:

“(2) concerns and trust around the COVID-19 vaccine (concerns and trust – specific attitudinal and risk-perception beliefs);”

Under 2.3 we now also state:

“The questionnaire was developed based on the available literature on COVID-19-vaccine hesitancy and vaccine hesitancy in general (Daly and Robinson, 2021; Dror et al., 2020; Neumann-Böhme et al., 2020; Quinn et al., 2019; Taylor et al., 2020) and further revised based on a qualitative pretest with students (data not published – in this pretest we asked for example about information needs and trusted resources). “

(See also #21)

4

Sample characteristics: although the student panel as a whole might be representative of all students at Maastricht University, the final sample (n=434) might not. The authors should address this in their discussion. For sure, the results are not to be generalized to the total population of Maastricht University Students (as is done on p11, line 301-302)

Thank you for this critical comment. We added in the discussion:

“Fourth, we used an already existing student panel for the data collection. Although the student panel is a representative of the university students and the response rate was relatively high for this study, there might some deviations in the responses.”

5

p5, lines 207: the authors might be more explicit on what is a ‘high’ correlation coefficient and what is considered ‘room for improvement’.

Our apologies for this confusion. We now added our criteria more explicitly:

“With that, all items that correlated significantly with intention and have room for improvement (we defined ‘room for improvement as having a score  less than 6), are potentially relevant as potential targets for future interventions.”

6

g., Results. Risk perception. Three of four significant correlations are small to medium. The strongest correlating item has limited room for improvement. Nonetheless they are all considered important targets for intervention.

See also #5. All determinants that are significantly correlated with intention and have a mean scores less than 6 were selected. Of course, at all times, the selection has to be done carefully. To make this more clear, we added to our discussion:

Additionally, the CIBER approach is helpful in eliminating irrelevant/not changeable determinants, but selection has to be done carefully at all times – sometimes, for example, it is needed to create interventions to keep a specific determinant at a certain high level.

7

CIBER plots. Since they are the most informative results of the study in terms of which determinant to choose for intervention, I would not place them in the Supplementary Materials. I think they are more informative than Table 2. Given the novelty of the CIBER plot approach, I feel that a thorough description of how to interpret the plot is warranted.

Thank you for this feedback. We elaborately described the CIBER approach and how to interpret the CIBER plots in the data analysis section (2.4). We now also added an extra reference as an example. Together with the changes made under #5 and #6, we hope the explanation is now sufficient.

Based on the reviewer’s suggestion, we now put the CIBER plots in the paper, and moved Table 2 to the supplementary files.

8

Discussion, p9, line 272. The authors state that intervention should also increase correlation with the behavior. What method do they suggest for achieving such strengthened association?

Our apologies for not being clearer: a very low correlation was found between a belief related to risk perception (i.e. physical consequences are not severe when contracting COVID-19) and intention to vaccinate. Ideally, this correlation should have been higher. Therefore, risk perception should be increased, either at the side of the actual risk, or the side of the actual consequences. A method to do this is “consciousness raising”. We now have clarified this further in text:

“for example “If I contract COVID-19, the physical consequences for me would be severe” had a mean that was relatively low, in combination with a relatively low correlation. Both should be higher (that is: ideally it is desired that people perceive COVID-19 as having severe consequences). One method for increasing risk perception (and the correlation with intention to vaccinate) is “consciousness raising” (either about the risk, or about the consequences).

9

Background characteristics. Are there relevant differences in uptake intention and determinants between Dutch and International students? I could imagine that for example trust in the government could differ. Is it known where the international students would be vaccinated (the Netherlands vs their home country)?

Thank you for this helpful feedback. We now did some additional analyses and did not find any differences between groups in vaccination intention, and no major differences between determinants. We added this to section 3.1:

“[…], no difference in vaccination intention was found between Dutch and International students (M=6.16 for Dutch students and M=6.23 for international students, p =.61). For the different underlying determinants, some determinants scored significantly different, but mean differences were for most small (Mean difference <.30 and all <.70  – Data not reported but can be found in the supplementary files)“

10

Intention. How many participants choose the “I don’t intend to take the vaccination” response option? How many answered “fully disagree”?

11 students fully disagreed to get vaccinated against COVID-19. We now added this to section 3.2:

“Of the 434 students, 348 (80.1%; score 6 – 7 [fully agree]) intended to get the COVID-19 vaccination when invited to do so (11 students fully disagreed to get vaccinated against COVID-19).

11

p5, lines 212-214. The additional items should be described in the method section.

Thank you for pointing out this. We now added the information below to the methods section:

“Additionally, three items were included to compare the COVID-19 vaccine with current vaccines in the National Immunization Program in relation to safety, effectiveness, and trustiness.”  

12

Discussion, p8, line 243: throughout the ms, the authors used the word ‘determinant’, but now they speak of ‘predictor’. Please be consistent in your wording.

Thank you. We followed your suggestion and throughout the manuscript, we changed the wording to prevent further confusion.

13

Discussion, p9, line 263. Do the authors have a suggestion on how to resolve whether the neutral scores on the comparison between COVID-19 and other vaccines might or might not be indicative of general vaccine hesitancy?

Thank you for this helpful thought. We now added to our discussion: “However, when students were asked whether the COVID-19 vaccines are equally safe, effective, and/or trusted compared to the current vaccines in the National Immunization Program, students mostly scored neutral, which might, or might not, be indicative of a general hesitancy about vaccines’ safety, effectiveness, and trustiness worldwide (Dubé et al., 2013). Future (potentially more qualitative) studies could help answering this question.”

14

Discussion, table 3. The meaning of the Change column is unclear to me (and layout is messy). If the determinants attitude, perceived norm and self-efficacy are not considered targets for improvement (p6, line 225), why are they marked with upward arrows? (what is the meaning of the arrows?)

Thank you for this feedback.

We now changed the page-orientation for this table and made sure that lay-out is improved.

In the change column, numbers indicate the mean scores, and the direction of the arrow refers to the direction of the intended change. We now added this explanation as footnote.

“* Numbers indicate the mean scores, and the direction of the arrow refers to the direction of the intended change.”

15

Discussion, p9, line 275-276: ‘reported elsewhere’: where?

Our apologies. The manuscript is in preparation. We changed this now to “data not published”.

16

Discussion, p9, intervention: Do I understand that an intervention was developed? has it been deployed/implemented/tested?

This intervention was developed and has (very) recently been accepted for publication. We now clarify this in this paper:

Based on this study, an actual intervention was developed that existed of a series of videos on a special website of the university on COVID-19 directed at students. The actual form was an interview by one student with, each time, an expert. The first part was about risk perception and worries and trust, with two experts in clinical microbiology, the second part on attitudes and perceived norms with two experts in health promotion/health psychology, and the third part was about perceived control was covered with clear online instructions how, where and when to get the vaccine, especially focused on international students. More information about the intervention development and lessons learned can be found in Ten Hoor et al., (2021).” 

17

Discussion, limitations. The statement that there might be other determinants is vague and nugatory. Can the authors make some theory-based or empirically informed suggestions?

Thank you for your feedback. We now added the following to our discussion:

“Systematic or scoping reviews compiling the theories used in the studies of COVID-19 vaccination or vaccination in general might be helpful for the identification of the determinants of vaccine intention and provide a roadmap for future vaccine studies.”

Reviewer 2 Report

This cross-sectional study aimed to examine university students’ intention to get the COVID-19 vaccine and identify the determinants of their behaviour. The association between the possible determinants and the outcome was investigated by means of a statistical system that simultaneously takes into account the correlation and the placement of the value of the determinant in the scale. The research project was well designed. The authors downplayed the limitations of the study in this project, which were conveniently discussed.

Author Response

Reviewer 2

This cross-sectional study aimed to examine university students’ intention to get the COVID-19 vaccine and identify the determinants of their behaviour. The association between the possible determinants and the outcome was investigated by means of a statistical system that simultaneously takes into account the correlation and the placement of the value of the determinant in the scale. The research project was well designed. The authors downplayed the limitations of the study in this project, which were conveniently discussed.

Thank you for your positive feedback.

Reviewer 3 Report

 The paper was a visualization of Covid-19 vaccination intention among youth, including students, using CIBER plots to identify the determinants of this intention. While the data are valuable enough to understand the vaccine intentions of young people, I recommend that this paper be accepted after the following three points are corrected and added to the paper.

1: A comparative study should first be conducted to better clarify the concerns with thevaccine against COVID-19 compared to other vaccinations (i.e., seasonal influenza vaccines, etc.). I would like to see a study on what makes it different from other vaccinations.

2: I would like you to present the sources of information that the students can get and how much they trust the information they get from those sources.

3: This survey is a survey of intentions toward the COVID-19 vaccine in the Netherlands. Please discuss fully that the results obtained here (other determinants that were positive and the most appropriate determinants to target) may be different in other areas.

Author Response

Reviewer 3

The paper was a visualization of Covid-19 vaccination intention among youth, including students, using CIBER plots to identify the determinants of this intention. While the data are valuable enough to understand the vaccine intentions of young people, I recommend that this paper be accepted after the following three points are corrected and added to the paper.

Thank you for your positive feedback and helpful suggestions.

1: A comparative study should first be conducted to better clarify the concerns with the vaccine against COVID-19 compared to other vaccinations (i.e., seasonal influenza vaccines, etc.). I would like to see a study on what makes it different from other vaccinations.

Thank you for this interesting suggestion. We agree that this would be interesting, but this is outside the scope of the current study. Based on Reviewer 1 (#2), we added some helpful references to our introduction (Larson et al, 2014; Xiao & Wong, 2020; Guidry et al, 2021).

2: I would like you to present the sources of information that the students can get and how much they trust the information they get from those sources.

Thank you. In a qualitative study (in preparation), we asked students about their information needs and trusted sources of information.

In our paper we added now:

“The questionnaire was developed based on the available literature on COVID-19-vaccine hesitancy and vaccine hesitancy in general (Daly and Robinson, 2021; Dror et al., 2020; Neumann-Böhme et al., 2020; Quinn et al., 2019; Taylor et al., 2020) and further revised based on a qualitative pretest with students (data not published – in this pretest we asked for example about information needs and trusted resources). “

(See also #3)

3: This survey is a survey of intentions toward the COVID-19 vaccine in the Netherlands. Please discuss fully that the results obtained here (other determinants that were positive and the most appropriate determinants to target) may be different in other areas.

Thank you for your critical comment. We addressed it 1) in the introduction:

“In a systematic review by Larson et al., (2014), an attempt was made to understand vaccine hesitancy and its determinants but answers remained inconclusive: they concluded that determinants of vaccine hesitancy are context-specific and varying across time, place and type of vaccine.”

And 2) by adding the following in our discussion:

“Third, this study was conducted in the Netherlands. As countries enforced varied regulations during the COVID-19 pandemic, selected relevant determinants may differ in other countries (see also Larson et al., 2014).”

Reviewer 4 Report

Dear Authors, Your review is up-to-date and focuses on an extremely topical issue in the field of autism. I applaud you for the rigorousness of your methodology. The results are clear and well described. The conclusions help the reader become aware of the factors predicting the availability of vaccines 

Author Response

Reviewer 4

23

Dear Authors, Your review is up-to-date and focuses on an extremely topical issue in the field of autism. I applaud you for the rigorousness of your methodology. The results are clear and well described. The conclusions help the reader become aware of the factors predicting the availability of vaccines 

Thank you.